# Endothelial Nitric Oxide Synthase in the Perivascular Adipose Tissue

**DOI:** 10.3390/biomedicines10071754

**Published:** 2022-07-21

**Authors:** Andy W. C. Man, Yawen Zhou, Ning Xia, Huige Li

**Affiliations:** Department of Pharmacology, Johannes Gutenberg University Medical Center, 55131 Mainz, Germany; wingcman@uni-mainz.de (A.W.C.M.); yawezhou@uni-mainz.de (Y.Z.); xianing@uni-mainz.de (N.X.)

**Keywords:** vascular function, obesity, nitric oxide, adiponectin, SIRT1

## Abstract

Perivascular adipose tissue (PVAT) is a special type of ectopic fat depot that adheres to most vasculatures. PVAT has been shown to exert anticontractile effects on the blood vessels and confers protective effects against metabolic and cardiovascular diseases. PVAT plays a critical role in vascular homeostasis via secreting adipokine, hormones, and growth factors. Endothelial nitric oxide synthase (eNOS; also known as NOS3 or NOSIII) is well-known for its role in the generation of vasoprotective nitric oxide (NO). eNOS is primarily expressed, but not exclusively, in endothelial cells, while recent studies have identified its expression in both adipocytes and endothelial cells of PVAT. PVAT eNOS is an important player in the protective role of PVAT. Different studies have demonstrated that, under obesity-linked metabolic diseases, PVAT eNOS may be even more important than endothelium eNOS in obesity-induced vascular dysfunction, which may be attributed to certain PVAT eNOS-specific functions. In this review, we summarized the current understanding of eNOS expression in PVAT, its function under both physiological and pathological conditions and listed out a few pharmacological interventions of interest that target eNOS in PVAT.

## 1. Introduction

Perivascular adipose tissue (PVAT) is a special type of ectopic fat depot that adheres to most large arteries and veins, small and resistance vessels, and microvessels of the musculoskeletal system [1]. The beneficial role of PVAT was first observed by Soltis and Cassis in the aorta of a Sprague–Dawley rat where that PVAT diminished agonists-induced vasocontraction in vitro [2]. Till now, PVAT has been shown to exert anticontractile effects in the blood vessels in both rodents and humans [3,4]. Similar to other adipose tissues, PVAT is an important endocrine tissue that secretes adipokines, hormones, growth factors, chemokine, reactive oxygen species (ROS), nitric oxide (NO), and hydrogen sulfide (H_2_S) [1]. As PVAT has a very close proximity to the vasculature, PVAT has been recognized as an active player in vascular physiology and pathology, and studies of PVAT in maintaining vascular homeostasis have been focused on in recent decades. Endothelial nitric oxide synthase (eNOS; also known as NOS3 or NOSIII) is an enzyme named after the cell type (endothelial cell) in which it was first identified. eNOS is well-known for its role in the generation of vasoprotective NO. To date, numerous studies using global eNOS-deficient mice have demonstrated the antihypertensive, antithrombotic, and anti-atherosclerotic effects of eNOS, which were mainly attributed to NO derived from the endothelium. Indeed, eNOS expression has been identified in both endothelial cells and adipocytes in PVAT and both contribute to the production of vascular NO and modulate vascular pathophysiology [5,6]. Although there are reviews discussing several aspects of PVAT, the functions of eNOS in PVAT have not been fully described. This review will address the current understanding of PVAT eNOS and propose possible roles of eNOS in PVAT for future directions.

## 2. What Is Special about PVAT?

There are three layers in the vascular wall of blood vessels, namely tunica intima, tunica media, and tunica adventitia. The inner layer, tunica intima, consists of a single layer of flattened, polygonal endothelial cells supported by a basal lamina of connective tissues. Tunica media is the middle layer that mainly consists of vascular smooth muscle cells (VSMCs), especially in arteries. Tunica adventitia is the strongest layer that contains connective tissues and elastic fibers [7]. Different from other adipose tissues, PVAT can be found outside the adventitia of the systemic blood vessels, including arteries and veins, small and resistance vessels, and microvessels in skeletal muscles. PVAT is absent in microvasculature and the cerebral vasculature [8,9]. There are no laminar structures or barriers between PVAT and the adventitia layer of blood vessels.

PVAT is a highly heterogenous tissue. In addition to stem cells, immune cells, and nerves, both white and brown adipocytes can be found in PVAT. White adipocytes mainly act as energy storage in the form of triglyceride [10], while brown adipocytes are more metabolically active and associated with thermogenesis [11]. There are regional phenotypic and functional differences among the PVAT in different locations of the vascular system [8,9]. Depending on its location on the vascular bed, PVAT can be white adipose tissue (WAT)-like (such as mesenteric PVAT), brown adipose tissue (BAT)-like (such as thoracic aortic PVAT) or mixed (such as abdominal aortic PVAT). Vascularization and innervation of these PVATs, as well as their adipokine profiles also highly vary [8,9,12,13,14], which could explain the local functional differences of PVATs. However, the morphology of PVAT in other species are currently less characterized than that in murine models.

Studies have shown evidence that as an anatomically separated adipose tissue, adipocytes from different PVATs may arise from unique progenitor cells, giving rise to its distinctive morphological and functional characteristics compared to other adipose tissues [15,16,17]. Nevertheless, the origins of adipocytes in the PVAT and the precise process of PVAT development are barely known. A recent study suggested that adipocytes in periaortic PVAT may partly originate from progenitors expressing smooth muscle protein 22-alpha (SM22α) [18]. Thoracic periaortic PVAT may present both SM22α+ and myogenic factor 5 (Myf5+) origins [19], whereas these progenitor cells are able to differentiate into uncoupling protein-1 (UCP-1) positive adipocytes in vitro [20]. A recent study has also shown that fibroblastic progenitor cells, but not VSMCs, are responsible for the adipogenesis of thoracic PVAT [21]. The origins of adipocytes in abdominal periaortic PVAT are less known. They may share, at least, similar developmental origins with SM22α+ and peroxisome proliferator-activated receptor gamma (PPARγ+) VSMCs, as the absence of PPARγ in the VSMCs resulted in a complete lack of abdominal periaortic PVAT development [22]. In the same study, Chang et al. also suggested that mesenteric PVAT may share a similar developmental origin with VSMCs, since the absence of PPARγ in VSMCs also resulted in a dramatic loss of mesenteric PVAT, while other adipose tissues were not affected [22]. Indeed, studies have also suggested that the developmental origins of mesenteric PVAT may be similar to the visceral adipose tissues [23,24]. Taken together, the lack of the discovery of unique cell markers makes the generation of PVAT-specific gene modified mouse models and the mechanistical study of PVAT function a challenging task.

## 3. What Is the Function of PVAT?

Since the first attention to the paracrine effects of PVAT on blood vessels [2], growing studies, from experimental animal models to clinical samples, have indicated that the cross-talk between PVAT and its connecting vessel plays a critical role in the physiological homeostasis and pathological changes of the cardiovascular system. The paracrine crosstalk between PVAT and its connecting vessel can actively regulate vascular inflammation and remodeling [23], while PVAT can also act as an endocrine organ to modulate multiple biological processes by releasing adipokines [25]. In 2002, using the physiological buffer in which PVAT from a healthy rat was incubated, Lohn et al. observed a direct relaxation in precontracted, isolated, PVAT-removed rat thoracic aorta [26]. They concluded the presence of transferable soluble substances from the PVAT that were released to the buffer and caused relaxation. It is currently known that PVAT is capable to synthesize and secrete substances via endocrine and paracrine mechanisms, including adipokines, growth factors, ROS, NO, and H_2_S [1]. The previous literature already explored the function of PVAT in detail [1,25,27,28,29,30]. Here, we briefly summarized the area of PVAT-derived adipokine production and vascular function regulation, and some novel findings of exosomes/extracellular vesicles (Figure 1).

Similar to any other adipose tissues, PVAT plays a critical role in vascular homeostasis via secreting adipokine, hormones, and growth factors [31]. These PVAT-derived factors include both pro-inflammatory and anti-inflammatory vasoactive molecules that modulate vascular inflammation and oxidative stress, vascular tone, and VSMCs proliferation and migration [9,32]. In various models of metabolic and cardiovascular diseases, including obesity, hypertension, and diabetes, the loss of anticontractile function of PVAT was observed [33,34,35]. PVAT becomes dysfunctional and the secretion of the PVAT-derived factors becomes imbalanced which could exert detrimental effects on vascular homeostasis and lead to vascular remodeling and arterial stiffening [28,36,37,38].

It is currently known that PVAT exerts anticontractile function on the adherent blood vessel through secretion of various PVAT-derived relaxing factors (PVRFs), previously known as the adventitia-derived relaxing factors (ADRFs) [39]. Potential PVRFs include leptin and adiponectin [40], H_2_S [41], hydrogen peroxide (H_2_O_2_) [42], prostaglandins [43,44], NO [45], and angiotensin (Ang) 1–7 [46]. In addition to PVRFs, recent studies have revealed that PVAT can secrete contracting factors (PVCFs) that modulate vasoconstriction [47,48,49]. Potential PVCFs include chemerin [50], calpastatin [51], 5-hydroxytryptamine (5-HT) [49], norepinephrine (NE) [52], AngII, and ROS [53]. Although the detailed mechanisms of how PVRFs and PVCFs exert their effects on the blood vessel remain unclear, it is hypothesized that PVAT modulates vascular functions through two distinct mechanisms: endothelium-dependent and endothelium-independent pathways [29,45]. These factors from PVAT may reach the endothelial layer of blood vessels by either direct diffusion or via vasa vasorum or small media conduit networks connecting the medial layer with the underlying adventitia [8,54,55] (Table 1). In addition, the same factors from PVAT can act as either PVRFs or PVCFs. For example, H_2_S and prostanoids in PVAT have anticontractile effects under normal conditions, while they can induce contractile responses under disease conditions [56].

A recent study has also shown that the anticontractile effects of PVAT can be attributed to its ability to uptake and metabolize vasoactive amines such as dopamine, NE, and 5-HT [62]. Monoamine oxidase A/B (MAO-A/B) catalyzes the oxidative deamination of vasoactive amines, while semicarbazide-sensitive amine oxidase (SSAO) catalyzes the generation of ammonia and H_2_O_2_. These two enzymes are present in PVAT, and the inhibition of these enzymes increased the NE-induced vasocontraction on vessel rings with PVAT [62]. PVAT can also prevent NE-induced vasocontraction, by taking up NE and preventing it from reaching the vessel wall [63].

In small arterioles, stepwise increase in blood pressure can induce vasoconstriction due to smooth muscle myogenic response, while this physiological function is absent in large arteries [64]. Until now, most of the in vitro pressure myograph studies about myogenic responses were performed in PVAT-denuded vessels. Therefore, there is an underlying question of whether PVAT may be involved in the regulation of myogenic responses. In resistance arteries with myogenic response, endothelial-derived hyperpolarization plays a more prominent role than NO in vasodilation [65]. Thus far, there has been no direct evidence on whether PVAT plays a role in myogenic response in vivo. Nevertheless, recent studies have shown the new function for PVAT in assisting stress-induced relaxation [66] and the presence of stretch sensitive, nonselective cation channel Piezo1 in PVAT [67]. These shed light on the possible function of PVAT in modulating myogenic responses.

Dysfunction of PVAT has also been linked to the development of atherosclerosis. Adipocytes and macrophages in PVAT can secrete various pro-inflammatory cytokines and adipokines including monocyte chemoattractant protein-1 (MCP-1), tumor necrosis factor alpha (TNF-α), leptin, adiponectin, omentin, etc. [68]. In obesity, inflammation in PVAT causes the phenotypic switch from anti-inflammatory to pro-inflammatory [69]. A recent study has also revealed that macrophage polarization in the PVAT is critically associated with coronary atherosclerosis [70]. M1 macrophages in the PVAT are positively correlated with a higher risk of coronary thrombosis and are correlated with plaque progression and destabilization. M2 macrophages in the PVAT are negatively correlated with increased arterial obstruction, calcification, necrosis, and decrease of the number of vasa vasorum in the adventitia layer [70]. Transplantation of PVAT from high-cholesterol diet-fed apolipoprotein E (ApoE) knockout mice to normal chow-fed ApoE-knockout mice resulted in a striking increase in atherosclerosis development [71]. These suggest that the inflammatory status of the PVAT is related to the progression of atherosclerosis.

Apart from the above secretory factors, adipocytes from PVAT also secrete different types of extracellular vesicles, including exosomes and microvesicles [72,73]. Exosomes are formed within the endosomal network and exocytosed by fusion with the plasma membrane, while microvesicles are directly formed from the plasma membrane. Extracellular vesicles are crucial regulators of vascular functions by transferring the enclosed biological messengers, including lipids, proteins, noncoding RNAs, and microRNAs (miRNAs) for intercellular communications [74]. Adipose tissues have been shown to produce circulating exosomal miRNAs, as a form of adipokine, to regulate gene expressions locally or distantly [75]. These miRNA-containing extracellular vesicles act as the agent for the crosstalk between adipose tissues and neighboring tissues, including endothelial cells, VSMCs, and macrophages [76,77,78]. In addition, the crosstalk between endothelial cells and adipocytes is modulated, at least partly, by the extracellular vesicles-mediated reciprocal trafficking of caveolin-1 (Cav-1) [79]. A recent study demonstrated that PVAT secretes encapsulated microRNAs, such as miR-221-3p, which can be taken up in neighboring VSMCs, and triggers an early vascular remodeling in vascular inflammation [72]. In another recent study, PVAT-derived exosomes were demonstrated to reduce macrophage foam cell formation through miR-382-5p- and bone morphogenetic protein 4 (BMP4)-PPARγ-mediated upregulation of cholesterol efflux transporters [76]. However, it is still unclear which cell types in PVAT generate these extracellular vesicles.

## 4. Current Proves of eNOS in PVAT

There are currently three isoforms of NO synthase (NOS), which is named by the cell types where they are first identified: neuronal NOS (nNOS or NOS1), inducible NOS (iNOS or NOS2), and eNOS (or NOS3) [77]. Vascular nNOS is expressed in perivascular nerve fibers and in the vascular wall, while the expression of iNOS is induced under conditions of inflammation and sepsis [77]. eNOS is primarily expressed, but not exclusively, in endothelial cells. All three isoforms of NO synthase catalyze the production of NO from L-arginine [80]. Under physiological conditions, eNOS is the main vascular source of NO, modulates vascular functions and confers protection against cardiovascular diseases.

In recent years, eNOS expression in other cell types has been demonstrated in vitro and in vivo. Indeed, eNOS expression has been found in dendrite cells [78], red blood cells [81], hepatocytes [80], as well as in adipocytes [6]. While the expression of iNOS in PVAT is only induced in pathological conditions [82], and the expression of nNOS in PVAT is controversial [83], the expression of eNOS in thoracic aortic PVAT has recently been demonstrated by various groups. Gene and protein expressions of eNOS in PVAT have been detected [6,84]. Using immunohistochemistry, eNOS can be stained in both adipocytes and endothelial cells of the capillaries and vasa vasorum in aortic PVAT [6,85]. Of the three isoforms of NOS, immunostaining of eNOS is the most abundant in PVAT of the saphenous vein, and eNOS activity is comparable in PVAT and the adherent vein [85]. In addition, in situ NO production in PVAT adipocytes can be directly detected by fluorescence imaging [13,86]. There is a high histological discrepancy of eNOS expression among the anatomical localizations of PVAT. Abdominal PVAT has been shown to have a lower eNOS expression compared with thoracic PVAT, while the eNOS expression remains the same along the vessel walls [13]. Indeed, unpublished data from our laboratory suggests a similar eNOS expression level of mesenteric PVAT and thoracic PVAT. Nevertheless, due to the highly heterogenous origins and compositions of different PVATs, detailed investigations of specific cell types that express eNOS in different PVATs is crucial.

## 5. What Are the Functions of eNOS in PVAT?

Unfortunately, due to the lack of PVAT-specific eNOS knockout animal models, the exact functions of eNOS in PVAT is relatively unclear. Most of the current knowledge about PVAT eNOS is based on evidence from studies using global eNOS knockout mice or mice with pathological conditions that leads to downregulation of PVAT eNOS. Here, we summarize current understanding of potential eNOS functions in PVAT under both basal and pathological conditions.

The first and most important function of eNOS in PVAT is, of course, to generate vasoactive NO. Previous studies with animal models have demonstrated that PVAT plays a crucial role in vascular NO production [1,6,29]. PVAT-derived NO can diffuse into the adjacent VSMC, stimulating soluble guanylate cyclase (sGC) and increasing the cyclic guanosine monophosphate (cGMP) level, which leads to the phosphorylation of large-conductance calcium-activated potassium channels in VSMC via protein kinase G, resulting in hyperpolarization and vascular relaxation [87,88]. In small arteries isolated from visceral fat of healthy individuals, basal vascular NO production is reduced after PVAT removal, which leads to an attenuated contractile response to L-NAME [89]. In PVAT-adhered, endothelium-denuded rat mesenteric arteries, inhibition of eNOS significantly enhances NE-induced contraction, indicating that eNOS in PVAT contributes to the vascular NO production, while the anticontractile effect of PVAT is, at least partly, independent of the endothelium [33,90]. In low-density lipoprotein receptor (Ldlr) knockout mice, the thoracic aortic PVAT shows significant upregulation of eNOS expression and NO production, which protects against impaired vasorelaxation to acetylcholine and insulin [84]. In a very recent clinical study, the authors demonstrated PVAT as a predominant source of NO in human vasculature in a no-touch saphenous vein grafts (NT-SVGs) coronary artery bypass model [91]. The study showed that PVAT, via NO production from eNOS, can induce vasorelaxation even in endothelium-denuded SVG. The above evidence suggests the protective role of PVAT eNOS in improving endothelial functions. Nevertheless, currently, there is a lack of detailed studies that are designed to compare the NO production and eNOS function among vascular components, such as the endothelium and PVAT.

In addition to direct modulation of vasodilation, PVAT-derived NO released toward the vascular lumen is a potent inhibitor of platelet aggregation and leukocyte adhesion [92]. PVAT has been shown to play a role in the inhibition of DNA synthesis, mitogenesis, and proliferation of VSMCs [93]. The inhibition of platelet aggregation and adhesion also protects VSMCs from exposure to platelet-derived growth factors. These confer the ability of PVAT to protect against the onset of atherogenesis and vascular remodeling in the adherent vessels. However, there is a lack of direct evidence of how PVAT eNOS and PVAT-derived NO act on atherogenesis and vascular remodeling.

Another important function of PVAT eNOS is to stimulate the expression of adiponectin, which is an important adipokine that contributes to vasodilation regulation, anti-inflammation, and inhibition of VSMCs proliferation and migration [36,94]. eNOS has been shown to regulate adiponectin synthesis in adipocytes by increasing mitochondrial biogenesis and enhancing mitochondrial function [95]. PVAT-derived adiponectin may regulate endothelial functions, partly by enhancing eNOS phosphorylation in the endothelium [96]. Indeed, the function of PVAT is determined by the browning and inflammation status. Mitochondrial biogenesis is involved in the browning of adipocytes [97]. Fitzgibbons et al. proposed that promoting the browning of PVAT might confer a protective effect to attenuate the development of vascular diseases [11]. PVAT eNOS may have a vital role in the mitochondrial biogenesis and browning of PVAT [98]. However, the detailed mechanisms underlying browning or thermogenesis of PVAT are barely known.

Apart from the functions of PVAT eNOS and NO mentioned above, NO is also known as an endogenously produced signaling molecule that regulates gene expression and cell phenotypes [99]. Currently, NO is known to regulate gene expression either by direct interaction with transcription factors or by post-translational modification of proteins. NO may mediate the transcriptional regulation of histone-modifying enzymes and modulate the activities and cellular localizations of transcription factors through the formation of S-nitrosothiols or iron nitrosyl complexes [100]. Additionally, NO may alter the cellular methylation, acetylation, phosphorylation, ubiquitylation, or sumoylation profiles of proteins and histones by modifying these enzymes [101]. Recent evidence has revealed the presence of S-nitrosylated (SNO) proteins in abdominal aortic PVAT [102]. For example, a reduced NO level results in the activation and cellular release of tissue transglutaminase (TG2), which is involved in vascular fibrosis and remodeling [103,104]. Normally, TG2 can be S-nitrosylated by NO, and is retained within the cytosolic compartment. Reduced bioavailability of NO leads to reduction of TG2 S-nitrosylation, which facilities its translocation to the extracellular compartment where it can induce crosslinking of extracellular matrix proteins and promote fibrosis [105]. Nevertheles, the complete nitrosylation profile of PVAT and vascular wall remains unclear. Identification of these SNO proteins could greatly enhance our understanding of the detailed function of PVAT eNOS and its derived NO.

A recent study has revealed the reciprocal regulation of eNOS and β-catenin [106]. eNOS and β-catenin are interactive partners. β-catenin is a membrane protein known to bind with eNOS to promote eNOS phosphorylation and activation, while this interaction facilitates the translocation of β-catenin to the nucleus and activates downstream gene transcription [106]. This suggests a potential role of eNOS as a ‘carrier’ protein to facilitate gene expression independent of NO production. In addition, another cobinding protein and negative regulator of eNOS, Cav-1, is expressed in both endothelial cells and adipocytes [107]. Cav-1 can regulate eNOS functions in PVAT [108], whereas eNOS-derived NO has been shown to promote caveolae trafficking [109]. These suggest that protein–protein interaction of eNOS may play a critical role in PVAT functions, such as the secretion of miRNA-encapsulated microvesicles.

## 6. PVAT eNOS under Pathological Conditions

Multiple studies with high-fat diet (HFD) and/or genetic manipulation models have reported the pathophysiological significance of PVAT eNOS in mediating vascular functions, inflammation, and other metabolic processes [6,29,110]. PVAT eNOS plays a crucial role in obesity-induced vascular dysfunction [1,28]. Indeed, endothelium-dependent, NO-mediated acetylcholine-induced vasodilation response has no significant changes in PVAT-removed aortas from HFD-fed mice compared with control mice, while vascular dysfunction of the thoracic aorta is only evident when PVAT is adhered [6,111]. Our group has also found evidence of PVAT eNOS dysfunction and eNOS uncoupling in HFD-induced obese mice [6]. Although an adaptive overproduction of NO from mesenteric PVAT was observed at the early phase of HFD-induced obesity in C57BL/6J mice [86], reduced eNOS expression was observed after long-time HFD feeding in the mesenteric PVAT of obese rats [33] and thoracic aortic PVAT of mice [112]. Either improving L-arginine availability [6] or restoring eNOS phosphorylation and acetylation [111] can ameliorate obese-linked vascular dysfunction. These suggest that obese-induced eNOS dysfunction in the PVAT can sigificantly reduce the vascular functions in the adherent vessels. In addition, basal NO production is reduced in small arteries from obese patients compared with nonobese controls, while this reduction in basal NO production is only evident in PVAT-adhered, but not in PVAT-removed arteries [89]. However, in HFD-fed ApoE knockout rat models of early atherosclerosis, Nakladel et al. demostrated an upregulation of eNOS in the inflammatory thoracic PVAT, which compensates severe endothelial dysfunction by contributing to NO production upon cholinergic stimulation [82]. Nevertheless, these results indicate that, under obesity-related metabolic diseases, PVAT eNOS may be even more important than endothelium eNOS in obesity-induced vascular dysfunction, which may be attributed to certain PVAT eNOS-specific functions [1,28,113].

The reduction of eNOS activity and PVAT function can be caused by the reduced L-arginine bioavailability and changes in post-translational modifications of eNOS in obese PVAT [28]. Deficiency in L-arginine is attributable to an upregulation of arginase in the PVAT of obese mice [6]. The upregulation of arginase reduces L-arginine bioavailability for NO production and leads to eNOS uncoupling in PVAT [114], while uncoupled eNOS produces superoxide and increases oxidative stress in PVAT [6]. Indeed, acylation, acetylation, S-nitrosylation, glycosylation, glutathionylation, and phosphorylation of eNOS have been reported and involved in the dynamic control of its activity in response to differrent physiologic and pathophysiologic cues [115]. Reduction in eNOS phosphorylation at serine 1177 residue and inhibition of Akt, an upstream kinase of eNOS, were observed in the thoracic aortic PVAT of obese mice [6]. Another important post-translational modification of PVAT eNOS involved in obesity-induced vascular dysfunction is acetylation [35,115]. eNOS has been reported to be constitutively acetylated at Lys 497 and Lys 507 [115], which inhibits the activity of eNOS. Deacetylation of eNOS by SIRT1 increases the enzymatic activity of the eNOS [116]. In our previous study, we observed an upregulation of eNOS acetylation in the thoracic aortic PVAT of obese mice [111]. O-GlcNAcylation is another post-translational modification of eNOS that influences its stability, activity and subcellular localization [115]. O-GlcNAcylation of eNOS in PVAT is increased in high sugar diet-fed rats as well as in hyperglycemic human patients, suggesting that O-GlcNAcylation of eNOS may be involved in high sugar diet-induced oxidative stress in PVAT [117]. Other modifications of eNOS and the resulting changes in eNOS functions have not been reported or investigated in PVAT in pathological models.

One of the mechanisms leading to eNOS dysfunction in PVAT is the dysregulation of leptin, adiponectin, and chemerin. HFD-induced obesity enhances the leptin level in PVAT which leads to the reduction of eNOS activity and NO production [86]. The reduction in PVAT eNOS activity and NO production in obesity can be partially attributed to the reduced expression of adiponectin in PVAT [88]. Adiponectin and eNOS have a bidirectional regulation. The decreased adiponectin from PVAT may also reduce endothelial eNOS activity in obesity [110,113,118]. In obesity, chemerin from PVAT contributes to the positive amplification of sympathetic nerve stimulation and thereby increases vascular tone [119], while chemerin in the vessel wall decreases the expression of the rate-limiting enzyme for tetrahydrobiopterin (BH4) biosynthesis, GTP cyclohydrolase I (GTPCH1), decreases eNOS activation and NO production, and promotes ROS generation [120,121]. Nevertheless, the regulation of PVAT eNOS by chemerin has not been investigated.

On the other hand, both aging and obesity might affect PVAT in a comparable manner [10]. Aging has been shown to attenuate the anticontractile effect of aortic PVAT and reduce the browning phenotype of PVAT in rats [122]. Aging can also promote obesity-induced oxidative stress and inflammation in PVAT, which in turn exacerbates the secretion of inflammatory factors from PVAT, and affects vascular remodeling in obese mice [123]. In addition, ROS production in PVAT is progressively increased during aging, which subsequently contributes to aging-related vascular injuries [122,124]. eNOS uncoupling has been demonstrated in aged vessels [125]; however, the changes in expression and uncoupling of eNOS in aged PVAT is totally unknown. Future studies are needed to examine eNOS expression and function during aging in related to aging-induced vascular complications.

## 7. Pharmacological Targeting of PVAT eNOS

As mentioned above, under obesity-related metabolic diseases, PVAT eNOS may be even more important than endothelium eNOS in obesity-induced vascular dysfunction. Therefore, restoring the function of eNOS in obese PVAT may effectively improve and normalize vascular functions. As many studies have focused on the pharmacological interventions targeting PVAT eNOS in obesity, different targets that regulate eNOS in PVAT have been detailled [28,126,127]. Here, we briefly summarize a few of interest.

SIRT1 is known as a class III histone deacetylase which also deacetylates nonhistone proteins and cytosolic molecules such as eNOS. SIRT1 deacetylates eNOS at lysine 494 and 504 in the calmodulin-binding domain of eNOS, resulting in the activation of eNOS [116]. Adipose tissue-specific-SIRT1 knockout mice have increased obesity-induced brown-to-white transition in PVAT in vivo, leading to impaired vascular reactivity [128]. The deficiency of PVAT SIRT1 may reduce PVAT browning by promoting local superoxide production and reducing adipokines production [128], which could be attributed to the inactivation of eNOS due to the constitutive acetylation of eNOS. In a very recent study, we demonstrated that the PVAT SIRT1 activity is reduced in obese mice despite an enhanced SIRT1 expression [35]. This resulted from the downregulation of NAD^+^-producing enzyme NAMPT, which leads to a reduced level of NAD^+^ and NAD^+^/NADH ratio in PVAT. The reduced SIRT1 activity is associated with an enhanced acetylation of eNOS in the PVAT [35]. In addition, activation of SIRT1 promotes mitochondrial biogenesis via the peroxisome proliferator-activated receptor-gamma and coactivator 1 alpha (PGC-1α) mitochondrial pathway in adipose tissues [129]. Moreover, SIRT1 is reported to regulate adiponectin secretion in adipocytes [130]. Resveratrol, a SIRT1 activitor, has been shown to improve PVAT functions [131,132], but the change in PVAT eNOS activity has not been studied. Nevertheless, these suggest a tight interplay between PVAT SIRT1 and eNOS in controlling the browning and inflammation status of PVAT, which mediates vascular functions (Figure 2).

The serine/threonine protein kinase Akt mediates the activation of eNOS, leading to increased NO production. Inhibition of Akt or mutation of the Akt binding site on eNOS at serine 1177 attenuates the phosphorylation of eNOS and prevents eNOS activation [133]. The standardized Crataegus extract WS^®^ 1442, with antioxidative properties, is known to enhance eNOS phosphorylation at the serine 1177 residue by stimulating Akt activity. Treatment with WS^®^ 1442 can restore the vascular functions in the PVAT-adhered aorta of obese mice without any effect on body weight or fat mass [27].

AMP-activated protein kinase (AMPK) is an important regulator of energy metabolism homeostasis and can activate eNOS via phosphorylation [134,135]. The activation of the AMPK/eNOS pathway in PVAT is responsible for its anticontractile function. Treating PVAT with various AMPK activators 5-Aminoimidazole-4-carboxamide ribonucleotide (AICAR), salicylate, metformin, methotrexate, resveratrol, or diosgenin was reported to increase phosphorylation of PVAT eNOS and improve PVAT functions in different studies [131,132].

Exercise training was shown to increase eNOS expression and eNOS phosphorylation in both vascular wall and PVAT, which is associated with increased adiponectin expression in PVAT [136]. Aerobic exercise training has been shown to promote the anticontractile activities of PVAT by upregulating the expression of antioxidant enzymes and decreasing oxidative stress in PVAT [126]. Aerobic exercise training also stimulates angiogenesis, which improves blood flow and reduces hypoxia and macrophage infiltration in PVAT [127]. In addition, exercise training induces browning and thermogenic response in rat PVAT, which is associated with increased eNOS expression and reduced oxidative stress [137]. Sustained weight loss also increases eNOS expression and improves NO production in PVAT from rats [33]. In rats fed with a high-fat/high-sucrose diet, exercise significantly increases adiponectin levels compared with nonexercised controls, which is associated with increased eNOS phosphorylation in PVAT [136]. Increased GTP cyclohydrolase 1 expression, which is involved in the production of BH4, an essential cofactor for NO generation from eNOS, was reported after exercise training in obese rat thoracic PVAT [138]. Moreover, bariatric surgery improved NO bioavailability in PVAT of small subcutaneous arteries from severely obese individuals [139]. These beneficial effects of exercise training and weight loss may be attributed to the restoration of eNOS activity (Figure 2).

## 8. Conclusions and Future Directions

PVAT has a unique role in the modulation of vascular functions due to its very close proximity to the vasculature. Important to note is also the significance of PVAT in modulating cardiovascular complications. In metabolic and cardiovascular diseases, adipose tissue dysfunction has a notable contribution to the associated vascular dysfunction. Recent evidence from different studies suggests that eNOS in PVAT, rather than eNOS in the vascular wall, plays a critical role in protection against obesity-induced vascular dysfunction (Figure 2). Conventional in vitro vascular experiments are mainly performed with PVAT-denuded vessels, which does not reflect the vascular function of in vivo conditions. In this regard, the study of PVAT functions and the unique role of eNOS in PVAT becomes extremely important for the investigation of metabolic and cardiovascular diseases and the research for pharmacological interventions. In order to have a better understanding of the unique role of eNOS in PVAT, there is an urgent need for a suitable animal model, i.e., a PVAT-specific eNOS knockout or overexpression mouse model. Nevertheless, due to the highly heterogenous origin and histological and functional variations among PVAT in different regions of the vascular bed, designing an ideal model for studying the specific functions of eNOS PVAT is a challenge. On the other hand, current understanding of eNOS functions in PVAT is based on the understanding of eNOS from endothelial cells, global knockout, or disease models. PVAT-specific gene knockout or overexpression animal models may help to answer the following questions:

oWhat is the exact of role of PVAT eNOS in PVAT functions?oWhat are the exact expression levels of eNOS in different regions of PVAT and/or in different cells in PVAT?oWhat is the relative contribution of endothelial eNOS and PVAT eNOS to vascular function under physiological and pathological conditions?oAre there any specific functions of eNOS in PVAT but not in endothelial cells?

## Figures and Tables

**Figure 1 biomedicines-10-01754-f001:**
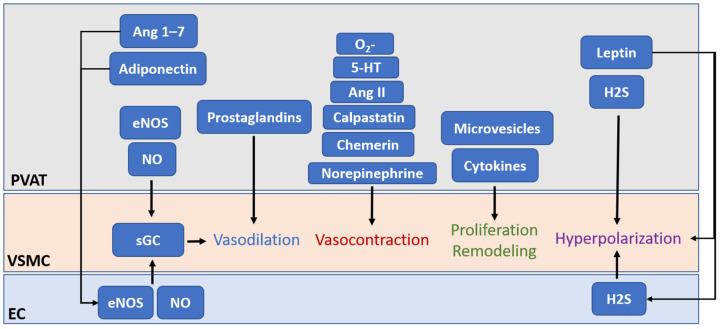
The crosstalk between PVAT and the vessel wall modulates vascular functions. PVAT releases vasoactive molecules, hormones, adipokines, and microvesicles. PVAT-derived relaxing factors (PVRFs) include leptin and adiponectin, hydrogen sulphide (H_2_S), hydrogen peroxide (H_2_O_2_), prostaglandins, NO, and angiotensin (Ang) 1–7. PVAT-derived contracting factors (PVCFs) include chemerin, calpastatin, 5-hydroxytryptamine (5-HT), norepinephrine (NE), AngII, and ROS. These factors from PVAT may reach the endothelial layer of blood vessels by either direct diffusion or via vasa vasorum or small media conduit networks connecting the medial layer with the underlying adventitia and modulate vasodilation and vasocontraction. PVAT plays a critical role in vascular homeostasis via secreting adipokine, hormones, and growth factors to modulate the proliferation of VSMCs. Adipocytes from PVAT also secrete different types of extracellular vesicles, including exosomes and microvesicles, which have also been shown to trigger early vascular remodeling in vascular inflammation. Under pathological conditions, PVAT becomes dysfunctional, and the secretion of the PVAT-derived factor becomes imbalanced which could exert detrimental effects on vascular homeostasis and lead to vascular remodeling and arterial stiffening.

**Figure 2 biomedicines-10-01754-f002:**
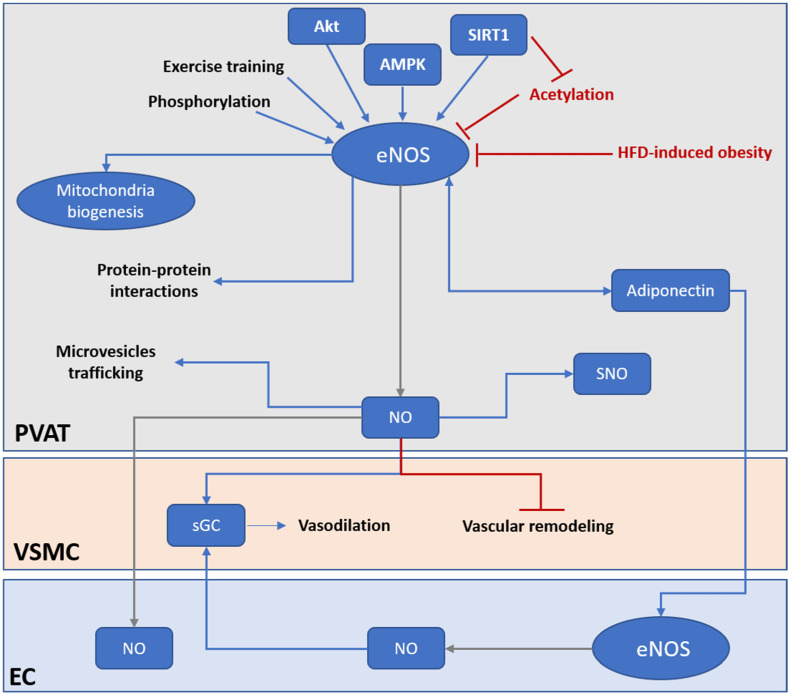
PVAT eNOS is an important modulator of vascular functions. Under HFD-induced obesity, the activity and expression of PVAT eNOS is significantly downregulated. PVAT eNOS may be even more important than endothelium eNOS in obesity-induced vascular dysfunction. Under normal condition, PVAT eNOS has multiple roles in regulating PVAT and vascular functions. PVAT eNOS can generate NO and regulate vasodilation via endothelium-dependent and endothelium-independent mechanisms. NO generated from PVAT eNOS can diffuse to the endothelium and activate EC, or directly activate sGC in the VSMC to evoke vasodilation. NO generated from PVAT eNOS can prevent vascular remodeling and stiffening via inhibiting VSMC proliferation and differentiation. PVAT eNOS is also responsible for modulating mitochondria biogenesis and browning of adipocytes in PVAT. In addition, NO generated from PVAT eNOS may regulate protein activities via SNO modification. Moreover, eNOS may, via protein–protein interactions and NO production, modulate miRNA-encapsulated microvesicles trafficking across PVAT. PVAT eNOS have a bidirectional regulation with adiponectin. Adiponectin is an important adipokine that modulates vascular functions via activating eNOS in both PVAT and EC. Current therapeutical strategies targeting PVAT eNOS include enhancing eNOS activity by phosphorylation, promoting deacetylation of eNOS via activation of SIRT1, activation of upstream kinase of eNOS (Akt, AMPK), and exercise training. AMPK, AMP-activated protein kinase; eNOS, endothelial nitric oxide synthase; EC, endothelial cell; HFD, high fat diet; NO, nitric oxide; PVAT, perivascular adipose tissue; sGC, soluble guanylyl cyclase; SNO, S-nitrosylation; VSMC, vascular smooth muscle cell.

**Table 1 biomedicines-10-01754-t001:** List of focused PVAT-derived factors.

PVAT-Derived Factors	Effects	References
Adiponectin	Relaxation	[40]
Angiotensin (Ang) 1–7	Relaxation	[46]
Angiotensin II (Ang II)	Contraction	[14,56,57]
Calpastatin	Contraction	[51]
Chemerin	Contraction	[50,54]
Hydrogen peroxide (H_2_O_2_)	Relaxation	[42,55]
Hydrogen sulphide (H_2_S)	Contraction	[41]
	Relaxation	[56,58]
Leptin	Relaxation	[57,59]
	Contraction	[51,60]
Nitric oxide (NO)	Relaxation	[45]
Norepinephrine (NE)	Contraction	[52]
Prostanoids		
-Prostaglandins	Contraction	[44,61]
-Prostacyclin	Relaxation	[22]
-Thromboxane	Contraction	[61]
Superoxide	Contraction	[53]
5-hydroxytryptamine (5-HT)	Contraction	[49]

## Data Availability

Not applicable.

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
