# Peer review of "Endothelial Nitric Oxide Synthase in the Perivascular Adipose Tissue"

_biomedicines, 2022, doi:10.3390/biomedicines10071754_

Round 1

Reviewer 1 Report

Deaer Authors

my compliments for your hard work.

Probably this paper needs more tables reporting exemplary data, this effort could give more informations in easy fashion.

Pleas report reference according tho the Authors' instructions.

Author Response

Reviewer 1:

Dear Authors

my compliments for your hard work.

Probably this paper needs more tables reporting exemplary data, this effort could give more informations in easy fashion.

Response:

We thank reviewer for the comment. A table summarizing the PVAT-derived factor has been added (page 4).

Pleas report reference according the the Authors' instructions.

Response:

We thank reviewer for the comment. The reference list has been revised according to the suggested template.

Reviewer 2 Report

Andy W.C Manet al. discussed the role of perivascular adipose tissue (PVAT)-derived nitric oxide on vascular functions. The current review summarized the current progress of research on PVAT and suggested that PVAT may play a role in vascular diseases. The manuscript is well-written and organized. I only have some minor concerns discussed below:

•          Small arterioles exert significant capability of vasoconstriction due to smooth muscle myogenic response in response to elevated blood pressure, but this physiological function is absent in the large artery vascular smooth muscle cells. Does PVAT play a role in this difference between small arteriole and large artery-like aorta? It would be great if the author can comment on this.

•          From vascular maturation perspectives, does the PVAT eNOS function differently at different ages of the animal/human?

•          Can the author comment on if PVAT contributes to the development of atherosclerosis or aneurysm?

Author Response

Reviewer 2:

Andy W.C Man et al. discussed the role of perivascular adipose tissue (PVAT)-derived nitric oxide on vascular functions. The current review summarized the current progress of research on PVAT and suggested that PVAT may play a role in vascular diseases. The manuscript is well-written and organized. I only have some minor concerns discussed below:

  • Small arterioles exert significant capability of vasoconstriction due to smooth muscle myogenic response in response to elevated blood pressure, but this physiological function is absent in the large artery vascular smooth muscle cells. Does PVAT play a role in this difference between small arteriole and large artery-like aorta? It would be great if the author can comment on this.

Response:

We thank reviewer for the suggestion. There is no clear answer to the question yet. Most of the current research about myogenic responses used PVAT-denuded vessels. It is conceivable that PVAT may play a role in myogenic response in vivo and there is yet no direct in vivo/in vitro evidence on whether PVAT plays a role. Nevertheless, a paragraph has been added to discuss this important issue (page 4 line 160-171).

  • From vascular maturation perspectives, does the PVAT eNOS function differently at different ages of the animal/human?

Response:

We thank reviewer for the comment. It is possible that PVAT eNOS functions differently during aging. However, there is currently a lack of studies on that. Nevertheless, a paragraph addressing PVAT in aging has been added (page 8-9, line 373-383).

Reviewer 3 Report

PVAT has recently been considered a relevant factor in vascular biology. The review prepared by  Andy W.C Man et al. summarises the current knowledge regarding the expression of eNOS in PVAT.

To improve the quality of the text and its usefulness to the readers, I would recommend to the authors only a minor revision of figure 1. This Figure should benefit from the typographical improvement to increase the readability of the text (especially the font size).

Author Response

Reviewer 3:

PVAT has recently been considered a relevant factor in vascular biology. The review prepared by Andy W.C Man et al. summarises the current knowledge regarding the expression of eNOS in PVAT.

To improve the quality of the text and its usefulness to the readers, I would recommend to the authors only a minor revision of figure 1. This Figure should benefit from the typographical improvement to increase the readability of the text (especially the font size).

Response:

We thank reviewer for the suggestion. Figure 1 has been revised accordingly to improve the readability (page 3).